# Mitral Regurgitation Severity Assessment after Transcutaneous Edge-to-Edge Mitral Valve Repair: Recommended Integration versus Volumetric Assessment Guidelines

**DOI:** 10.3390/jcm12196347

**Published:** 2023-10-03

**Authors:** Efrat Shekel, Mony Shuvy, Haim Danenberg, David Planer, Dan Gilon, David Leibowitz, Ronen Beeri

**Affiliations:** 1Heart Institute, Hadassah University Medical Center, Jerusalem 9112001, Israelplaner@hadassah.org.il (D.P.); dangi@ekmd.huji.ac.il (D.G.); oleibo@hadassah.org.il (D.L.); 2Cardiology Institute, Shaare Zedek Medical Center, Jerusalem 9103102, Israel; monysh@gmail.com; 3Heart Center, Wolfson Medical Center, Holon 5822012, Israel; danen040@gmail.com

**Keywords:** mitra regurgitation, Mitralclip, mitral TEER, echocardiography, residual MR, color Doppler

## Abstract

**Objectives:** This article aims to evaluate the accuracy of the color-Doppler-based technique to evaluate residual mitral regurgitation post TEER. **Background:** The evaluation of residual mitral regurgitation (MR) post-mitral transcutaneous edge-to-edge repair (mitral TEER) is a critical determinant in patients’ outcomes. The common methods used today, based on the integration of color flow Doppler parameters, may be misleading because of the multiple jets and high velocities created by the TEER devices. **Methods:** Patients undergoing TEER at Hadassah hospital were recruited between 2015 and 2019. Post-procedural MR was evaluated using the integrated qualitative approach as recommended by the guidelines. In addition, the MR volume for each patient was calculated by subtracting the forward stroke volume (calculated by multiplying the LVOT area with the velocity time integral of the LVOT systolic flow) from the total stroke volume (calculated by the biplane Simpson method of discs). We compared the two methods for concordance. **Results:** Overall, 112 cases were enrolled. In 55.4% of cases, the volumetric residual MR was milder than the MR severity assessed by the guidelines’ recommended method. In 25.1%, the MR severity was similar in both methods. In 16.2%, the MR severity was worse when calculated using the volumetric method (*p*_Value_ < 0.001, Kappa_measure of agreement_ = 0.053). The lower residual MR degree using the volumetric approach was mostly observed in patients classified as “moderate” by the integrated approach. **Conclusions:** MR severity after TEER is often overestimated by the guideline-recommended integrative method when compared with a volumetric method. Alternative methods should be considered to assess the MR severity after mitral TEER.

## 1. Introduction

Mitral regurgitation (MR) is the most common valve disease and the second common indication for valvular surgery in the United States and Europe [1,2]. In the past few years mitral transcutaneous edge-to-edge repair (mitral TEER) is an important tool in the treatment of patients with severe mitral regurgitation (MR) who are not eligible for surgery [3]. Patients with a suspicion of MR undergo different test to conclude the presence of MR and its severity. The main tool used for this purpose is color flow Doppler (CFD), which allows a jet assessment using transthoracic echocardiography (TTE). This method is commonly used to assess the degree of MR before the procedure [4].

To assess the presence and severity of MR using TTE and CFD, specialists have described several techniques. The main conclusion was to consider the origin of the regurgitation jet and its width, the orientation of the jet in the receiving chamber, and the flow convergence into the regurgitation orifice [5]. But, when discussing this issue in the context of mitral TEER it becomes much more complicated. Although CFD is the main tool in use for valvular diseases in general, a post-mitral TEER evaluation is different and much more challenging [6].

However, for efficiency purposes and, more importantly, due to the lack of a gold standard method, the CFD technique is widely used. In conclusion, the MR is qualitatively assessed using those parameters from echocardiography post TEER as well, using the same technique used for native, untreated valves [5].

It has been suggested before that this technique in not suitable for use post-TEER valve fitting. Furthermore, even the current guidelines regarding the assessment of post-TEER MR mentions the different challenges in assessing MR post TEER, especially CFD limitations in this field [6]. There are different reasons that can explain the challenges mentioned. An important one is the fact that TEER often creates multiple systolic orifices [7]. The current quantitative methods (such as the use of PISA or vena contracta diameter) are limited by the presence of the devices, with multiple regurgitant orifices [8]. Restricting the flow orifice of a given regurgitant jet increases its velocity, and thus may give the erroneous impression of a more severe jet due to the increased aliased jet area with flow entrainment. Nonetheless, assessing the jet area in the left atrium by color Doppler remains the mainstay of MR assessment after TEER [5].

When trying to understand the limitation described, we found an important observation that has been noted and described in different articles to support our theory. TEER has shown a prognostic benefit in terms of hospitalization and mortality, with no significant difference between patients with none, 1+, or 2+ grade of residual MR [9]. In addition, hemodynamic methods, such as left atrial pressure changes, showed a prognostic benefit, regardless of the color-Doppler-based residual MR severity [10]. The findings described may suggest that the evaluation of residual MR is wrong, meaning that the residual MR post TEER is milder than it appears and assessed by the CFD technique, that is why we see a much larger effect clinically than sonographically.

Therefore, our hypothesis is that MR severity is overestimated by the guideline-recommended integrative method after TEER.

## 2. Methods

We performed a retrospective observational study of collected data and echocardiography studies of patients who underwent TEER at the Hadassah Hebrew University Medical Center between 2015 and 2019. The study protocol was approved by the institutional ethical review board and all patients provided informed consent for data collection.

Mitral TEER was performed according to standard of care. Through a transfemoral-venous transseptal access, the MitralClip device was placed on the valve leaflets according to the place of the regurgitation. When the placement was satisfactory the device was locked in place [3]. In several cases, the patient required multiple clips to achieve the desired effect. The procedure was declared as finished when the maximum effect, according to the doctors performing the procedure, was achieved.

Transthoracic echocardiography (TTE) was performed to assess the MR severity after TEER using standard protocols, meaning a CFD-based qualitative assessment. To assess the effect properly, and because of the progressive nature of MR, we included in our study only patients who underwent post-mitral TEER transthoracic echocardiography up to a week after the procedure, we excluded patients with TTE performed over a week from the procedure.

MR severity was assessed by integrating parameters from the echocardiography, including the jet area in relation to the left atrial area, the width of the jet at its origin (vena contracta diameter), and the size of the proximal isovelocity surface area (PISA). This multiparametric visual evaluation was performed by an experienced echocardiographer and is compatible with methods acceptable by the echocardiography community [11]. By integrating these measures, the MR severity was graded on a 6-point scale: none, mild, mild to moderate, moderate, moderate to severe, and severe [5].

In parallel, a quantitative volumetric calculation of the residual MR volume (RVol) was performed according to the method described in Foster et al. [4]. Each patient underwent a qualitative assessment as described before, and a quantitative assessment as follows: we calculated the left ventricle end-diastolic volume (LV EDV) and left ventricle end-systolic volume (LV ESV) using the biplane method of discs (modified Simpson’s method) [12,13]. The total stroke volume (SV) was calculated by subtracting the ESV from the EDV. The forward SV was calculated by multiplying the LV outflow tract area (diameter measured in the zoom mode and used to calculate area) by the left ventricle outflow tract velocity time integral (LVOT VTI) obtained using pulsed-wave Doppler. The mitral regurgitant volume (MR RVol) was calculated by subtracting the forward SV from the total SV.

A brief description of the calculation is presented below:EDV − ESV = Total Stroke Volume (SV)
LV Outflow tract area = (Outflow tract diameter/2)^2^ × π
LV Outflow tract area × LVOT VTI = Forward SV
Total SV − Forward SV = Regurgitation Volume (RVol)

We divided the patients into groups of MR severity by the calculated MR RVol according to the European Society of Echocardiography Guidelines [11]:None: Less than 5 mL.
Mild: 5 mL ≦ RVol ≧ 29 mL.
Mild to moderate: 30 mL ≦ RVol ≧ 44 mL.
Moderate to severe: 45 mL ≦ RVol ≧ 59 mL.
Severe: 60 mL and more.

The qualitative visual estimation of MR is categorized on a scale of 6 grades while the calculated scale is divided into 5 grades. Therefore, to achieve a better statistical concordance, 2 parallel analyses were performed: the first referred to all patients (24 patients in total) categorized as “moderate” by the visually estimated scale as part of the “mild to moderate” group. The second included these patients as part of the “moderate to severe” group (Figure 1).

For that reason, two sets of results will be presented from now on, a lighter assumption, calculating patients categorized “moderate” as “mild to moderate”, and a stricter assumption, calculating them as “moderate to severe”.

**Statistical methods:** The baseline clinical characteristics and demographic data of the patients were analyzed using a Mann–Whitney test. A McNemar–Bowker test was performed to assess the level of concordance between the qualitative MR grade and the quantitative grade. A Kruskal–Wallis Test was performed on variables related to post-procedural cardiac function. The variables taken into consideration were the number of clips placed, left ventricle (LV) dysfunction degree, mean post-procedure MV gradient, and clinical improvement on the NYHA scale. All of the parameters are presented in Table 1.

## 3. Results

A total of 160 TEER procedures were performed during the period of the study. Of these, 6 cases who underwent an emergency procedure and 36 patients who passed away prior to this study’s initiation were excluded. We also excluded 6 patients with missing data (Figure 2).

The patients were excluded as described to avoid any distractions regarding their general medical condition and to avoid difficulties in achieving relevant data. As the aim of this study is above all to hypothesize the benefits of TEER and to hopefully describe a better effect then seen before, we analyzed the clearest indications and the most common and straightforward procedures. Emergency procedures have fewer clear indications and are usually performed as a last resort, and thus, were excluded for those reasons. In addition, patients who died after the procedure lacked information and data in our database and were excluded as well for this reason.

A total of 110 patients were included in this study. Two of these patients underwent the procedure twice, with a year or more between procedures, thus 112 procedures were analyzed (Figure 2). The patients’ characteristics are shown in Table 1.

The MR grades are described in Figure 1, divided into qualitative grades and quantitative grades as shown.

Taking this information as the data, a McNemar–Bowker test was performed to assess the level of concordance between the two grade scales. We performed the same test twice: once considering the “moderate” group as “mild to moderate”, and the second time considering the “moderate” group as “moderate to severe”. This was performed for the reasons described before (see Section 2).

When the “moderate” MR grade was analyzed with the “mild to moderate” group, the statistical analysis reduced the MR severity in 55.4% (62) of the patients when comparing the qualitative to the quantitative grades. In 27.7% (31), the MR severity did not change, and in 17.1% (19) the MR severity was higher according to the quantitative method (*p* < 0.001, Kappa = 0.074) (Figure 3).

When the “moderate” grade on the qualitative approach was recorded as “moderate to severe”, the statistical analysis reduced the MR degree for 59% (66) of patients when the volume was calculated quantitatively. In 25.1% (28) of patients, the MR severity did not change, and in 16.2% (18), the MR severity was higher according to the calculated volume (*p* < 0.001, Kappa = 0.053) (Figure 4).

We would like to emphasize that the main group who benefited from this technique was the mid-range grades, meaning the moderate group and below, who showed the best improvement in their grade of MR severity.

No statistically significant associations were identified using the Mann–Whitney non-parametric test regarding age, gender, diabetes, hypertension, smoking, coronary artery disease, or MR etiology for either one of these approaches. This means that even when taking into consideration other medical conditions of the participants, the results are still statistically significant.

A Kruskal–Wallis Test was performed on the number of clips placed, left ventricle (LV) dysfunction degree, mean MR gradient degree, and improvement in NYHA scale (see Table 1). None of the analyzed parameters showed a statistically significant association with either of the approaches. This means that the cardiac parameters did not affect the benefits and improvement in the MR degree. All groups benefited from the quantitative assessment regardless of their cardiac function.

In conclusion, we demonstrated a consistent line of results showing a reduction in MR severity when calculating the RVol as opposed to assessing it qualitatively.

## 4. Discussion

In this study, we evaluated the accuracy of the echocardiographic technique used to assess residual MR after mitral TEER. We found that the RVol after TEER is overestimated when the integrated, guideline-recommended CFD-based method is utilized. These findings were consistent among the different groups of patients with different clinical characteristics.

Trying to explain the inaccuracy of the CFD-based method, we hypothesized several ideas that may explain our findings.

There are several factors known to affect the CFD jet appearance in MR which possibly make this technique unsuitable for use in post-TEER patients [5]. Factors that may change the MR jet appearance include technical features (such as the chosen Nyquist limit), the direction of the jet (with underestimation of highly eccentric jets), the flow rate (determined also by the LV systolic function) and velocity (determined also by the MV orifice). In Ref. [5], enhancing our hypothesis, the authors show that the flow continuity principle determines that flow, being a product of the area of an orifice multiplied by the velocity of the fluid, is constant [12]. Thus, when TEER creates several small orifices, it increases the velocity of the blood flowing through these orifices. These smaller orifices result in a larger color flow area due to higher velocities with aliasing, not necessarily reflecting a larger regurgitation volume. The direction of the jet also impacts the image. A centrally directed jet may look larger than an eccentric one alongside the wall of the atrium [14]. While different defects in the valve leaflets, such as prolapse or perforation, may affect the direction of the jets, TEER devices may change the jet direction and influence the severity interpretation.

Another finding worth mentioning is that the main discrepancies in MR severity assessments were in patients with a moderate grade or lower residual MR. Most of the patients had a lower residual MR grade using the volumetric technique when compared to the integrative technique. In the “severe” group, the grades did not change significantly. This trend is logically understandable since a “severe” MR has a blunt appearance that is hard for an experienced echocardiographer to miss. A severe MR is less sensitive to visual artifacts and is a much clearer image. By contrast, a diagnosis of “mild” or “mild to moderate” MR is much harder to differentiate by eye. That is where the quantitative assessment is important.

This trend is also compatible with prognostic outcomes in other studies. As described in Kal et al. [9], the major prognostic difference was between grade 2+ on the one hand, and the 3+ and 4+ grades on the other hand. While severe residual MR (3+ and 4+ grades) presented the expected clinical manifestation, moderate MR (2+ grade) had the same clinical outcome as “none” or mild MR. This suggests a possible inaccuracy in the 2+ and 1+ MR grades [9].

This inaccuracy is also seen when comparing the integrative MR assessment technique with hemodynamic indexes. For example, Taramasso et al. showed that pulmonary vein flow and MV gradients were better predictors for TEER procedural success when compared with the guideline-derived integrative MR assessment technique [15]. This may be another indication for the inaccuracy of the CFD-based method for evaluating residual MR post TEER.

We believe in seeking a more accurate method for evaluating residual MR. The connection between decreased residual MR and prognostic factors was originally described in the EVEREST II study [16]. This connection has been observed in additional studies since [17]. Therefore, we believe that an accurate evaluation of the residual MR is necessary for understanding the true effect and benefits of mitral TEER. An effect that is shown in a reduction in symptoms and a decreasing NYHA score. A correct assessment of the benefits of mitral TEER can potentially increase its use and indications.

We, therefore, suggest that additional approaches, less reliant on color Doppler, should be considered for the evaluation of post-TEER residual MR.

Hemodynamic parameters such as pulmonary venous flow, which has shown a good concordance with prognostic outcomes, immediately after TEER are an option [15,18]. The increasing use of three-dimensional imaging and multiplanar reconstruction can significantly improve the accuracy of color-flow-based methods. Measuring the 3D vena contracta area (which localizes with the effective regurgitant orifice area (EROA)) on the 3D color Doppler image, allows the measurement of eccentric or even multiple orifices with relative accuracy. Multiplying this area by the velocity time integral of the MR flow allows direct calculation of the regurgitant volume [19]. Furthermore, newer equipment can even measure the EROA automatically based on a 3D image, making this method even more practical and efficient [14]. Such 3D imaging methods include 3D echocardiography or even cardiac MRI, although MRI is still under research and not yet incorporated in the guidelines for this purpose, mainly because of the artifacts in the imaging created by the clips [6]. In this study, our patients did not undergo cardiac MRI, so the effects and benefits of MRI in this setting cannot be discussed here.

There are several limitations to this study.

First, this study is a small, single center, retrospective study, performed only on elective, relatively low-risk patients. Larger multicenter studies are needed to verify these findings.

Second, the technique used to measure the volumes used to calculate the RVol is not ideal. We used a 2D image to calculate 3D values and presumed an ideal space for the measurements. This might impact the accuracy of these measurements. In addition, we measured the values manually, exposing them to the possibility of human error. Furthermore, when calculating the volume from a 2D image, the value calculated is raised to the power of three, thus enhancing the influence of even small inaccuracies. With that said, it is important to note that this method, inaccurate as it may be, is used in the well-established and widely used biplane method of discs (modified Simpson’s method) [12,13].

Inaccurate as this method be, in the absence of a gold standard method, this method, with its limitations, is acceptable.

In conclusion, this study shows an inconsistency in the evaluation of post-TEER residual mitral regurgitation. Our study suggests that the current guideline-recommended CFD-based methods systematically overestimate residual MR after TEER.

It is important to acknowledge the fact that the opposite conclusion is also a possibility. Meaning that the quantitative method used in this study is underestimating the severity of MR.

With that said, other studies, as mentioned before, support the overestimation theory, and that is our suspicion as well. But further studies are required to verify our result.

As residual MR post TEER has prognostic consequences [17], additional studies are required, utilizing a more exact evaluation of the MR severity, such as 3D echo-based methods. Our findings require further prospective studies, with a larger number of patients, to assess the prognostic implications.

## Figures and Tables

**Figure 1 jcm-12-06347-f001:**
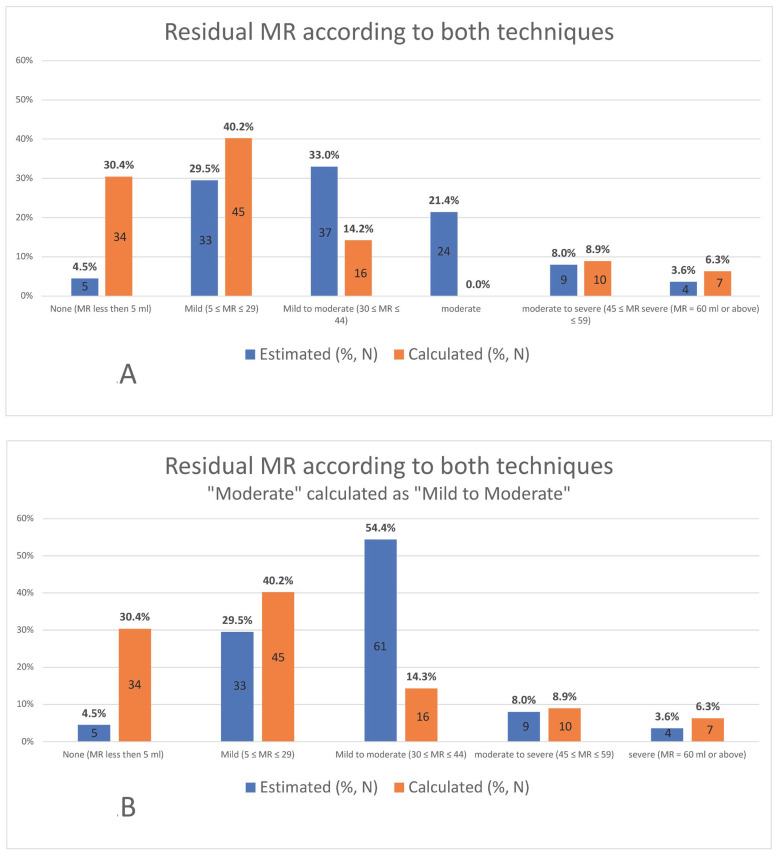
**Division of patients according to grades of residual MR in both methods.** The division with data unprocessed (**A**); the division when “moderate” group processed as part of “mild to moderate” (**B**); and the division when “moderate” group processed as part of “moderate to severe” (**C**). ”Estimated” refers to the use of color Doppler for the evaluation and “calculated” refers to the calculation according to the quantitative values measured from the echocardiography scan.

**Figure 2 jcm-12-06347-f002:**
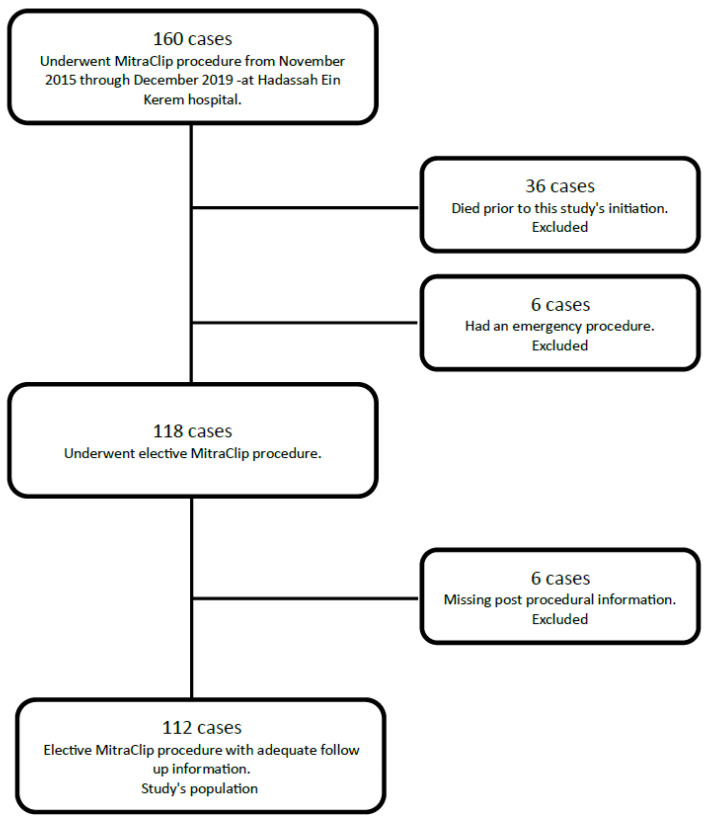
Patient selection process.

**Figure 3 jcm-12-06347-f003:**
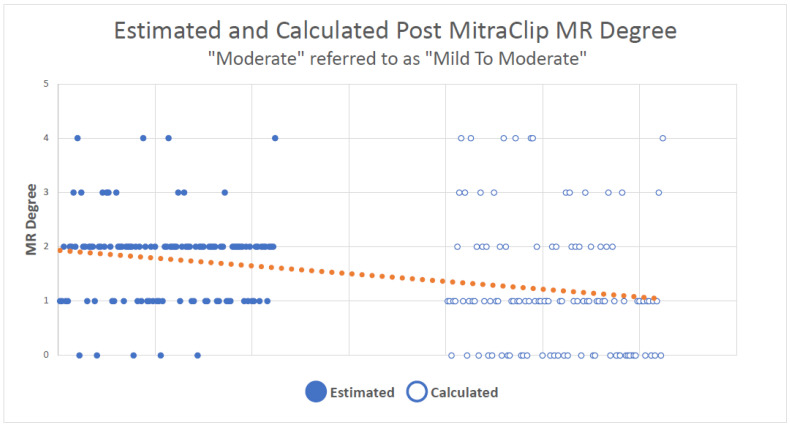
**The trend when comparing both methods with “moderate” group processed as “mild to moderate”**. Each dot represents a patient in the study. Full dots represent the grade according to the estimation method and the blank dots represent the grade according to the calculation. The linear graph shows the overall trend of lowering the grade.

**Figure 4 jcm-12-06347-f004:**
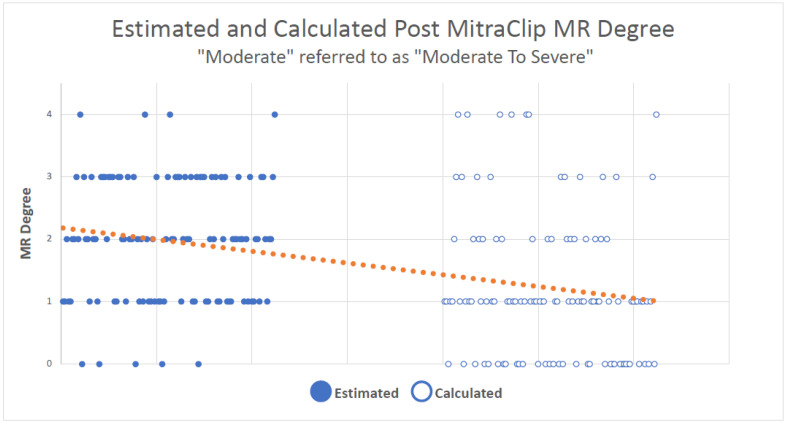
**The trend when comparing both methods with “moderate” group processed as “moderate to severe”**. Each dot represents a patient in the study. Full dots represent the grade according to the estimation method and the blank dots represent the grade according to the calculation. The linear graph shows the overall trend of lowering the grade.

**Table 1 jcm-12-06347-t001:** Baseline Patient Characteristics.

Age (Years) (Age range)	74 ± 11 (44–93)
Gender (%)	Male: 73 (65.2%)
	Female: 39 (34.8%)
Diabetes (%)	30 (26.8%)
Hypertension (%)	101 (90.2%)
Hyperlipidemia (%)	91 (81.3%)
Smoking (%) Tables:	None: 63 (56.3%)
	Current: 17 (15.2%)
	Past: 27 (24.1%)
	No information: 5 (4.4%)
Coronary artery disease (%)	62 (55.4%)
MR etiology (%)	Functional: 66 (58.9%)
	Degenerative: 39 (34.8%)
	Mixed: 4 (3.6%)
	No information: 3 (2.7%)
Number of clips placed in the procedure (%)	1: 34 (30.4%)
	2: 61 (54.5%)
	3: 15 (13.4%)
	No information: 2 (1.7%)
Estimated LV dysfunction according to Echocardiography (%)	Normal: 43 (38.4%)
	Mild: 11 (9.8%)
	Moderate: 17 (15.2%)
	Severe: 37 (33%)
	No information: 4 (3.6%)
Mean mitral valve gradient (%)	<5 mmHg: 81 (72.3%)
	≧5 mmHg: 29 (25.9%)
	No information: 2 (1.8%)
Degree of improvement on NYHA scale (%)	0 4 (3.6%)
	1: 47 (42%)
	2: 48 (42.9%)
	3: 6 (5.4%)
	No information: 7 (6.1%)

## Data Availability

Not applicable.

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
