# Peer review of "Mitral Regurgitation Severity Assessment after Transcutaneous Edge-to-Edge Mitral Valve Repair: Recommended Integration versus Volumetric Assessment Guidelines"

_jcm, 2023, doi:10.3390/jcm12196347_

Round 1

Reviewer 1 Report

Dear Authors,

Thank you for submitting your article: “Mitral Regurgitation Severity Assessment after Transcutaneous Edge-to-Edge Mitral Valve Repair: Guidelines-Recommended integration Versus Volumetric Assessment” to Journal of clinical medicine.

It is an interesting study, but I miss discussion including the new echo tools like True View and also to compare with results from patient specific silica models (Caballero et al 2022 compbiomed). The references are also in general old, please add some more recent, this also makes it difficult to compare to Guidelines.

It would be great if you specify TEER,  only 1st gen of MC? Not Pascal?

You state that residual MR has a significant prognostic consequence, though you did not discuss that very much, please add. Maybe include something on MitraScore? 

Author Response

attached a file with our notes. 

Thank you so much

Reviewer 2 Report

I’ve read with interest the manuscript titled „Mitral Regurgitation Severity Assessment After Transcutaneous Edge-to- 1 Edge Mitral Valve Repair: Guidelines-Recommended Integration VersusVolumetric Assessment by E. Shekel and associates.

            They compare the integrative approach to evaluating MR after Transcutaneous Edge-to-Edge Repair (TEER) recommended by the current guidelines with a quantitative method, based on a volumetric method and convincingly demonstrate that the former overestimates the degree of residual mitral regurgitation (MR). I have the following comments to make:

1.    They didn’t specifiy the aortic gradient of the patient cohort. This value can impact total forwad stroke measurement

2.    The simple volumetric method used by the authors (manually measuring the end-diastolic volume, the endsistolic volume and the total LVOT outflow volume), involving three manual measurements, has a three-fold potential for human error, and therefore is abandoned even for quantitative MR meaurements involving a single orifice (as found in organic MR), in favor of PISA

3.    They partially acknowledge this fact in the Limitations subsection of the manuscript

4.    Similary to PISA for unique-orifice MR, 3D PISA for multiple jets has less room for error and is recommended for quantitative echographic assessement.

5.    The conclusion that the current guidelines recommended integrative approach overestimates residual MR, could be more accurately rephrased that the volumetric method used by the authors underestimates residual MR after TEER

Author Response

Dear reviewer, 

Here below you can find an attached file with our notes. 

regards

Reviewer 3 Report

The authors compared guideline and volumetric assessments of residual MR after M-TEER for MR and concluded that guideline assessments overestimated the severity of residual MR. Although this study contains hot topics and has the potential to provide useful information in daily clinical practice, the following points should be reexamined.

1 Lack of gold standard method.

2 The timing of echocardiography is not clear. (About 20% of patients died before study entry)

3 Why is the following guideline not used? J Am Soc Echocardiogr. 2019 Apr;32(4):431-475.

4 Describe actual echocardiographic images and typical cases where the two methods discordance.

5 There are many items in Table 1 that do not add up to 112 cases, which is the number of eligible cases, and need to be confirmed

6 The order of Figures 1 and 2 are reversed.

Author Response

Dear reviewer,

hereby attached a file with all our notes regarding your review. 

Thank you and best regards

Round 2

Reviewer 3 Report

Thank you for submitting your manuscript for review. After a careful examination, there are several key points that need further clarification and attention;

1. MRI is frequently employed as the gold standard when assessing the comparative severity of mitral regurgitation. Could you provide specific reasons for not employing it in your study? Understanding this will shed light on your methodology and selection process.

2. The timeline for post-treatment assessment of mitral regurgitation is crucial for an accurate interpretation of results. Different timings can indeed affect the outcomes, and this potential variability may serve as a limitation to your study. Can you specify when the assessment was performed?

3. Case presentation concerns.

The color Doppler image presented for the 55-year-old case appears to represent the diastolic phase rather than the systolic phase. Could you confirm this?

The tracing line, as delineated using the Simpson method, seems to inaccurately represent the LV's apex. Please review this section for accuracy.

The left ventricular outflow tract (LVOT) diameter is vital for the calculation of stroke volume measurement. However, the reviewer noticed that the LVOT was not measured in the zoomed mode. This omission may influence the validity of your results, and we advise addressing this in your manuscript.

Author Response

Dear reviewer, 

Thank you for your review. 

Hereby attached a file with all of our answers (in Pink) next to your comments for clarity. 

Best regards!
